# Doxorubicin blocks proliferation of cancer cells through proteolytic activation of CREB3L1

**Bray Denard, Ching Lee, Jin Ye\***

Department of Molecular Genetics, University of Texas Southwestern Medical Center, Dallas, United States

**Abstract** Doxorubicin is used extensively for chemotherapy of diverse types of cancer, yet the mechanism through which it inhibits proliferation of cancer cells remains unclear. Here we report that doxorubicin stimulates de novo synthesis of ceramide, which in turn activates CREB3L1, a transcription factor synthesized as a membrane-bound precursor. Doxorubicin stimulates proteolytic cleavage of CREB3L1 by Site-1 Protease and Site-2 Protease, allowing the $NH_2$-terminal domain of CREB3L1 to enter the nucleus where it activates transcription of genes encoding inhibitors of the cell cycle, including *p21*. Knockdown of CREB3L1 mRNA in human hepatoma Huh7 cells and immortalized human fibroblast SV589 cells conferred increased resistance to doxorubicin, whereas overexpression of CREB3L1 in human breast cancer MCF-7 cells markedly enhanced the sensitivity of these cells to doxorubicin. These results suggest that measurement of CREB3L1 expression may be a useful biomarker in identifying cancer cells sensitive to doxorubicin.

## Introduction

Doxorubicin (Adriamycin) is used widely to treat diverse types of cancer, yet its effectiveness is hampered by the existence of drug-resistant cancer cells. The reason for drug resistance is unclear mainly because the mechanism through which doxorubicin inhibits proliferation of cancer cells is not completely understood. Doxorubicin has been proposed to exert its cytostatic action through intercalation into DNA and production of free radicals (*Gewirtz, 1999*). However, these mechanisms are unlikely to be clinically relevant as the concentration of doxorubicin required to produce these effects is much higher than that achievable in patients (*Gewirtz, 1999*). Inhibition of topoisomerase II by doxorubicin at clinically achievable concentrations leads to DNA breaks, but a consistent relationship between DNA strand breaks and the cytostatic action of the drug has not been demonstrated (*Gewirtz, 1999*). Thus, the mechanism through which doxorubicin inhibits cell proliferation remains unclear.

In addition to blocking cell proliferation, doxorubicin induces renal fibrosis in mice by stimulating production of collagen (*Lee and Harris, 2011*). The dual ability of doxorubicin to block cell proliferation and to induce collagen expression caught our attention inasmuch as we recently showed that both responses can be activated by a transcription factor called cAMP response element-binding protein 3-like 1 (CREB3L1, also known as OASIS) (*Denard et al., 2011*). CREB3L1 belongs to a family of transcription factors synthesized as transmembrane precursors (*Omori et al., 2002*) and activated through a process designated as Regulated Intramembrane Proteolysis (RIP) (*Brown et al., 2000*). The transcription factor domain of CREB3L1 is located in the $NH_2$-terminal 374-amino acids that project into the cytosol (*Figure 1A*). The COOH-terminal domain of 124 amino acids projects into the lumen of the endoplasmic reticulum (ER) (*Figure 1A*). Viral infection triggers the RIP of CREB3L1, which undergoes two sequential cleavages mediated by Site-1 protease (S1P) and Site-2 protease (S2P) (*Denard et al., 2011*). The S1P-catalyzed cleavage at the luminal side is a prerequisite for the S2P-catalyzed

**\*For correspondence:**
jin.ye@utsouthwestern.edu

**Competing interests:** The authors have declared that no competing interests exist

**eLife digest** Cancer is a broad term to describe over 200 diseases that are caused by cells proliferating in an out-of-control manner. Cell replication and division are normally very tightly regulated, and as cells become old, damaged or mutated, they are either repaired or undergo programmed cell death (apoptosis). However, if defective cells continue to replicate, the resulting clusters of abnormal cells can become cancerous.

With so many different types of cancer, there is no 'magic bullet' to cure all of them. Many cancer therapies are targeted, relying on drugs that block the spread of cancer by interfering with specific molecules involved in the growth and progression of certain tumors. However, the fact that diseased cells replicate faster than normal cells in many forms of cancer makes it possible to use non-specific drugs, such as doxorubicin, to treat tumors when targeted therapies are not available.

Doxorubicin can induce DNA breaks in a variety of different cancers by inhibiting the activity of topoisomerase II but a consistent relationship between the inhibition of this enzyme and the blocking of cell proliferation has not been established. This lack of understanding of the mechanism through which doxorubicin inhibits cell proliferation makes it difficult to identify cancer patients who are most likely to benefit from doxorubicin treatment.

Denard et al. have now shown that doxorubicin blocks cell replication by cleaving a transcription factor called CREB3L1. This latest work builds on previous work in which they showed that cleavage of this transcription factor can inhibit the replication of cells infected with hepatitis C virus. It has been known since 2000 that CREB3L1 is a membrane protein with one end inside the lumen of the endoplasmic reticulum, and the other end (which is terminated with an $NH_2$ group) in the cytosol of the cell. When CREB3L1 is cleaved, the $NH_2$-terminal domain travels into the nucleus of the cell, where it drives the transcription of genes that suppress the cell cycle. Denard et al. clearly show that doxorubicin triggers the cleavage of CREB3L1 by stimulating the production of ceramide molecules. Thus, It might be possible, with further research, to use CREB3L1 as a biomarker to identify tumors that are suitable for treatment by doxorubicin.

intramembrane cleavage that releases the $NH_2$-terminal domain of the protein from membranes, allowing it to drive transcription of genes that suppress cell proliferation such as *p21* (***Denard et al., 2011***). Nuclear CREB3L1 also activates genes required for assembly of the collagen matrix, including *collagen 1α1* (***Denard et al., 2011***). These dual activities prompted us to hypothesize that doxorubicin functions by stimulating proteolytic activation of CREB3L1.

In the current study, we determine that doxorubicin induces proteolytic activation of CREB3L1, and this cleavage is required for doxorubicin to inhibit proliferation of cancer cells. We further demonstrate that doxorubicin-stimulated production of ceramide is required for RIP of CREB3L1. These results suggest that CREB3L1 expression may be a useful biomarker in identifying cancer cells sensitive to doxorubicin.

## Results

To analyze proteolytic activation of CREB3L1, we fractionated human hepatoma Huh7 cells (***Nakabayashi et al., 1982***) into membrane and nuclear fractions, and used an antibody reacting against the $NH_2$-terminal domain of CREB3L1 (***Denard et al., 2011***) to examine the cleavage of CREB3L1 through immunoblot analysis. In the absence of doxorubicin, CREB3L1 existed as the full length precursor (~80 kDa) in membranes and the cleaved nuclear form of CREB3L1 (~55 kDa) was barely detectable (***Figure 1B***, lane 1). Treatment with doxorubicin markedly raised the amount of the nuclear form of CREB3L1 (***Figure 1B***, lane 2). The amount of membrane protein calnexin and nuclear protein lysine-specific demethylase 1 (LSD1) was not altered by doxorubicin treatment (***Figure 1B***). Doxorubicin may increase the amount of nuclear CREB3L1 through stimulation of CREB3L1 precursor cleavage or inhibition of nuclear CREB3L1 degradation, which was reported to be carried out by proteasomes (***Murakami et al., 2009***). To determine whether doxorubicin inhibits degradation of nuclear CREB3L1, we transfected Huh7 cells with a cDNA encoding $NH_2$-terminal fragment of CREB3L1 resembling the cleaved nuclear form of the protein (pCMV-CREB3L1(Δ381-519)) (***Denard et al., 2011***).

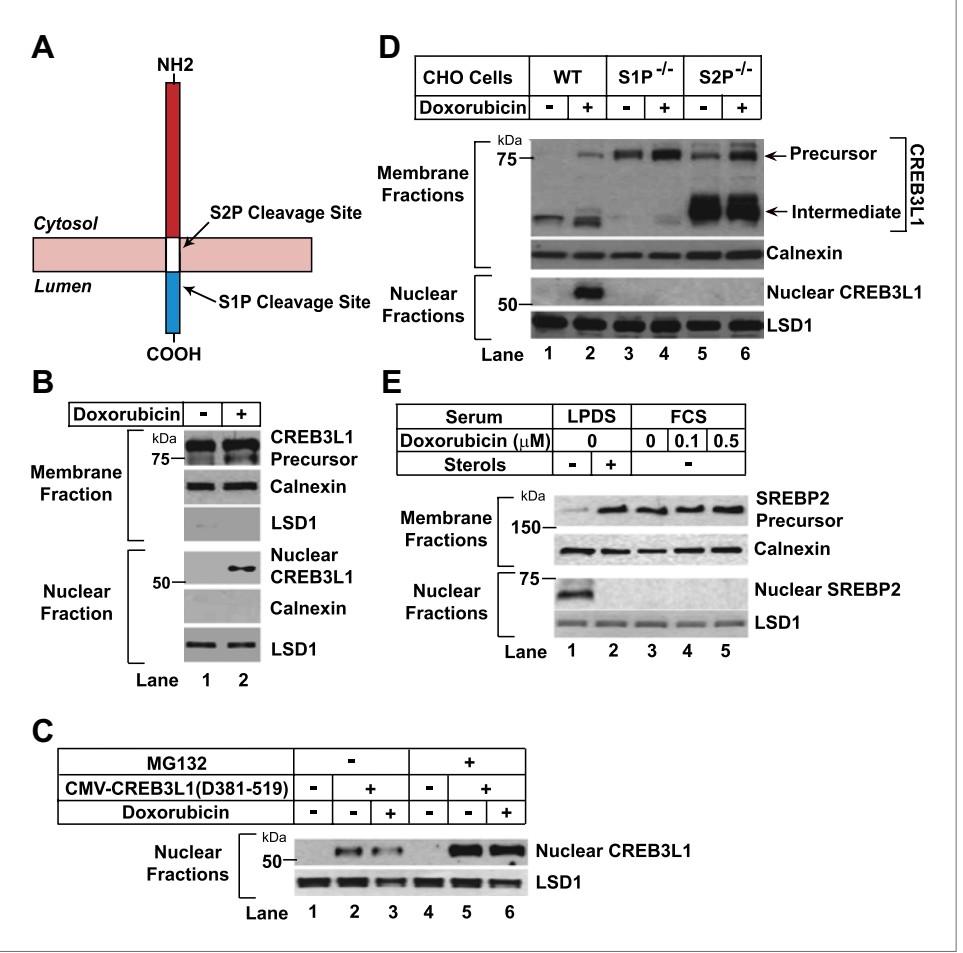

**Figure 1**. Doxorubicin stimulates RIP of CREB3L1. (**A**) Schematic diagram of CREB3L1. (**B**),(**D**) On Day 0, Huh7 cells (**B**) or wild type and mutant CHO cells (**D**) were seeded at 4 × 10⁵ cells per 60 mm dish. On day 1, cells were treated with 500 nM doxorubicin. On day 2, 24 hr after the treatment, the cells were separated into nuclear and membrane fractions, and analyzed by immunoblot with antibodies directed against CREB3L1, calnexin and LSD1. (**C**) On day 0, huh7 cells were seeded at 1 × 10⁵ cells per 60 mm dish. On day 1 they were transfected with pCMV-CREB3L1(Δ381-519) (0.1 µg per dish) as indicated. On day 2, they were treated with 500 nM doxorubicin as indicated. On day 3, 24 hr after the treatment, the cells were treated with 10 µM MG132 for 2 hr as indicated. Nuclear fraction of the cells was then analyzed by immunoblot analysis with antibody reacting against CREB3L1 and LSD1. (**E**) On day 0, CHO-7 cells were seeded at 2 × 10⁵ cells per 60 mm dish. On day 1, some cells were changed into sterol-depleting medium (medium A containing 50 µM compactin, 50 µM mevalonate, and 5% lipoprotein deficient serum [LPDS]) with or without supplementation of sterols (1 µg/ml 25-hydroxycholesterol and 10 µg/ml cholesterol). Other cells were changed into normal medium (medium A supplemented with 5% fetal calf serum [FCS]) containing the indicated concentrations of doxorubicin. On day 2, 24 hr after the treatment, the cells were separated into nuclear and membrane fractions, and analyzed by immunoblot with antibodies directed against SREBP2, calnexin and LSD1.

The amount of transfected nuclear form of CREB3L1 was not affected by doxorubicin (*Figure 1C*, lanes 2 and 3). However, this amount was increased in cells treated with the proteasome inhibitor MG132 (*Figure 1C*, lanes 5 and 6), suggesting that overexpression of the transfected protein did not overwhelm the machinery that degrades nuclear CREB3L1. These results suggest that doxorubicin does not stabilize nuclear CREB3L1. Thus, doxorubicin appears to increase nuclear CREB3L1 by stimulating proteolysis of its precursor.

To determine whether doxorubicin-stimulated cleavage of CREB3L1 was catalyzed by S1P and S2P, we analyzed the cleavage in mutant Chinese Hamster Ovary (CHO) cells deficient in S1P or S2P (*Rawson et al., 1997, 1998*). In wild type CHO cells, doxorubicin stimulated cleavage of CREB3L1 to

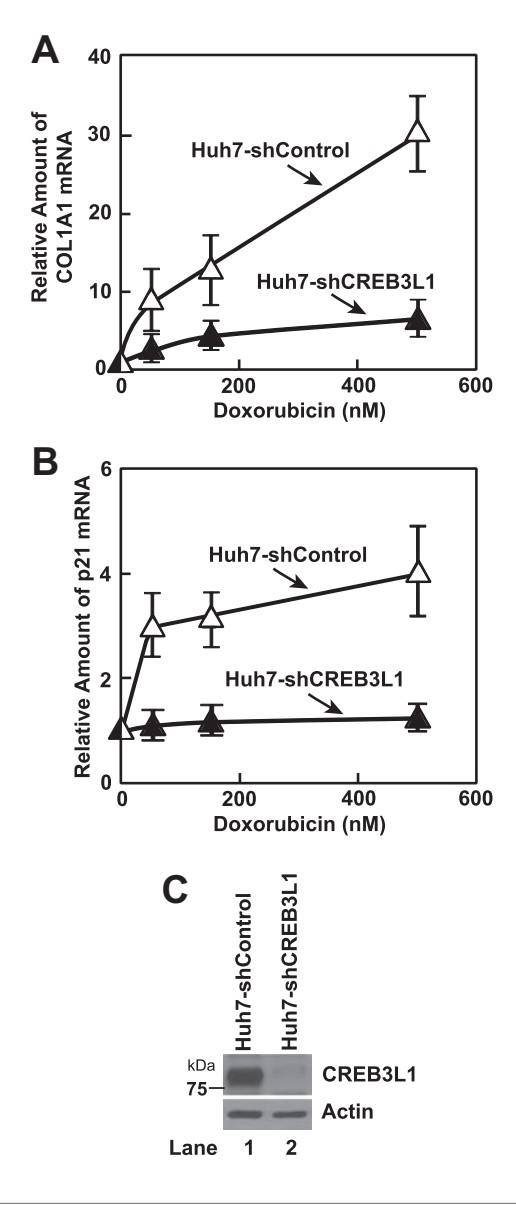

**Figure 2**. Doxorubicin induces transcription of genes activated by CREB3L1. (**A**),(**B**) On day 0, indicated cells were seeded at $3 \times 10^5$ cells per 60 mm dish. On day 1, the cells were treated with the indicated concentration of doxorubicin. On day 2, 24 hr after the treatment, some of the cells were harvested for quantification of p21 mRNA through RT-QPCR (**B**). On day 4, 72 hr after the treatment, the rest of the cells were harvested for quantification of *collagen 1α1* (COL1A1) mRNA through RT-QPCR (**A**). (**A**),(**B**) The value of each mRNA in cells that were not treated with the drug is set to 1. (**C**) Immunoblot analysis of CREB3L1 in indicated cells.

produce the nuclear form (*Figure 1D*, lane 2). In contrast, doxorubicin failed to produce the nuclear form of CREB3L1 in mutant cells deficient in either S1P or S2P (*Figure 1D*, lanes 4 and 6). In wild type CHO cells, we also detected in the membrane fraction a cleaved fragment with a molecular weight similar to that of the nuclear form (*Figure 1D*, lanes 1 and 2). This fragment was absent in cells deficient in S1P (*Figure 1D*, lanes 3 and 4) but dramatically elevated in cells deficient in S2P (*Figure 1D*, lanes 5 and 6). These findings suggest that this membrane-bound fragment is the intermediate form of CREB3L1 that was cleaved by S1P but not by S2P. Similar cleavage intermediates were observed in earlier studies of SREBP-2, a prototypes of RIP substrates, in mutant CHO cells deficient in S2P (*Rawson et al., 1997*; *Ye et al., 2000*). SREBP-2 was cleaved in sterol-depleted CHO cells (*Figure 1E*, lane 1) to activate genes required for cholesterol synthesis and uptake (*Brown and Goldstein, 2009*). However, this cleavage was not activated by doxorubicin (*Figure 1E*, lanes 4 and 5). Thus, doxorubicin appears to specifically induce proteolytic activation of CREB3L1.

An alternative approach to determine the effect of doxorubicin on proteolytic activation of CREB3L1 is to analyze the effect of the compound on expression of target genes activated by CREB3L1. In Huh7 cells transfected with a control shRNA (Huh7-shControl), doxorubicin induced the expression of *collagen 1α1* and *p21* (*Figure 2A,B*), both of which were shown to be direct targets of CREB3L1 (*Murakami et al., 2009*; *Denard et al., 2011*). In Huh7 cells stably transfected with a shRNA targeting CREB3L1 (Huh7-shCREB3L1) (*Denard et al., 2011*) in which expression of CREB3L1 was drastically reduced (*Figure 2C*), induction of these genes was markedly blunted (*Figure 2A,B*).

Inasmuch as CREB3L1 was required for doxorubicin to induce expression of *p21*, a well-characterized inhibitor of the cell cycle (*Sherr and Roberts, 1999*), we determined whether CREB3L1 was also required for doxorubicin to inhibit cell proliferation. For both untransfected Huh7 cells and those transfected with the control shRNA (Huh7-shControl), doxorubicin completely blocked their proliferation at a concentration between 50 and 150 nM (*Figure 3A*). This concentration of doxorubicin also resulted in maximal

cleavage of CREB3L1 in Huh7 cells (*Figure 3B*). For Huh7-shCREB3L1 cells, doxorubicin at concentrations up to 500 nM failed to block their proliferation (*Figure 3A*). These concentrations of doxorubicin were not enough to trigger apoptosis of Huh7 cells, which became apparent only when the cells were treated with 5 µM of the compound (*Figure 3C*). To rule out the off-target effects of the

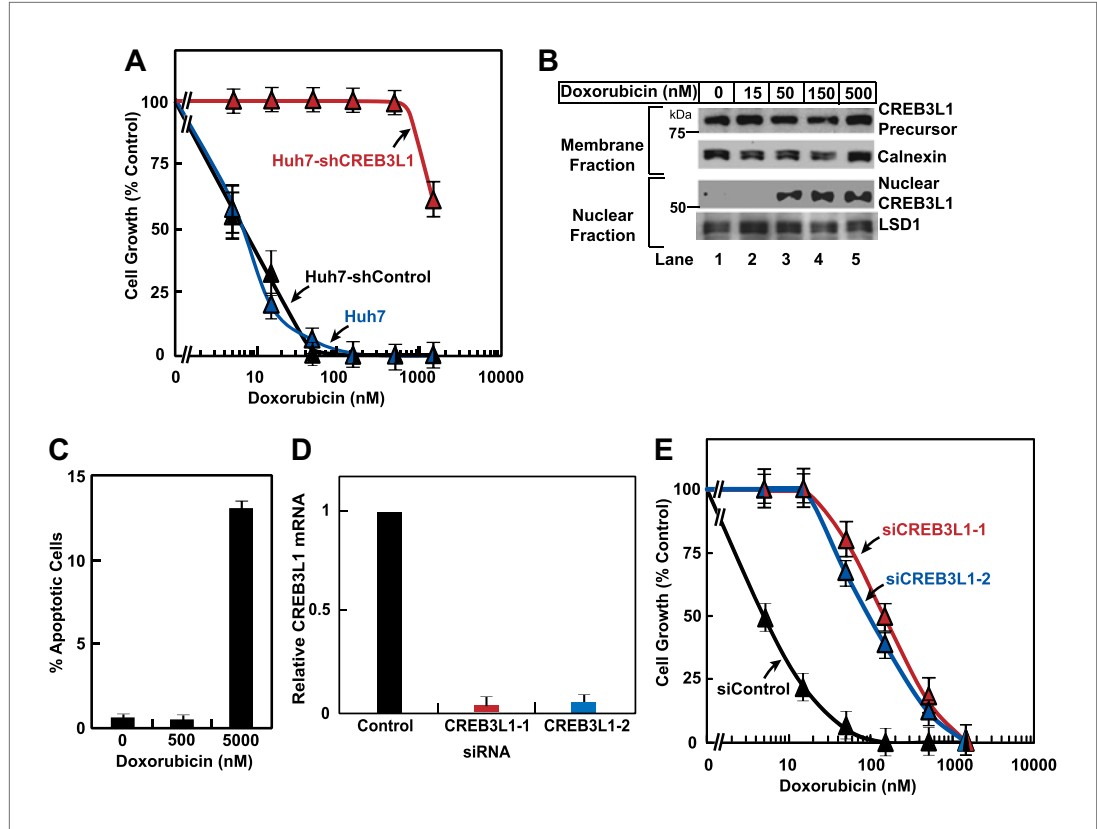

**Figure 3**. CREB3L1 is required for doxorubicin to suppress proliferation of Huh7 cells. (**A**) On day 0, indicated cells were seeded at $1.5 \times 10^5$ cells per 60 mm dish. On day 1, they were treated with the indicated concentrations of doxorubicin. On day 3, 48 hr after the treatment, the cells were quantified to determine cell proliferation. The number of cells just prior to the drug treatment and after treatment with no drug for 48 hr is set to 0% and 100%, respectively. (**B**) Huh7 cells treated with the indicated concentrations of doxorubicin were analyzed as described in *Figure 1B*. (**C**) On day 0, Huh7 cells were seeded at $4 \times 10^5$ cells per 60 mm dish. On day 1, cells were treated with the indicated concentrations of doxorubicin. On day 3, 48 hr after the treatment, cells were harvested to determine the percentage of the cells that underwent apoptosis through TUNEL assay. (**D**),(**E**) On day 0, Huh7 cells were seeded at $1 \times 10^5$ cells per 60 mm dish. On day 1, the cells were transfected with indicted siRNAs. On day 2, the cells were treated with indicated concentrations of doxorubicin. On day 4, 48 hr after the treatment, some of the cells were harvested for quantification of CREB3L1 mRNA by RT-QPCR (**D**), while the others were used for determination of cell proliferation as described in Figure 3A (**E**). (**A**),(**C**),(**D**),(**E**) Results are reported as mean ± S.E.M. of three independent experiments.

shRNA, we also transfected Huh7 cells with two distinct siRNA targeting regions of CREB3L1 that is different from that targeted by the shRNA. Transfection with these siRNA knocked down CREB3L1 mRNA by more than 90% (*Figure 3D*), and the treatment also rendered Huh7 cells more resistant to doxorubicin (*Figure 3E*).

If proteolytic activation of CREB3L1 is required for doxorubicin to inhibit cell proliferation, then the amount of CREB3L1 expressed in cancer cells may determine their sensitivity to doxorubicin. To test this hypothesis, we analyzed SV589 cells, an immortalized line of human fibroblasts (*Yamamoto et al., 1984*), and MCF-7 cells, a line of human breast cancer cells (*Soule et al., 1973*). Compared to Huh7 cells, expression of CREB3L1 was higher in SV589 cells and lower in MCF-7 cells (*Figure 4A*). The sensitivity of the cells to growth inhibition by doxorubicin followed the order of CREB3L1 expression (*Figure 4B*). Similar to Huh7 cells, knockdown of CREB3L1 by two duplexes of siRNA targeting different regions of CREB3L1 in SV589 cells (*Figure 4C*) made them more resistant to doxorubicin (*Figure 4D*). Since MCF-7 cells expressed very little CREB3L1, we used these cells to study the effect of CREB3L1 overexpression on sensitivity to doxorubicin. We stably transfected MCF-7 cells with a plasmid encoding CREB3L1 and

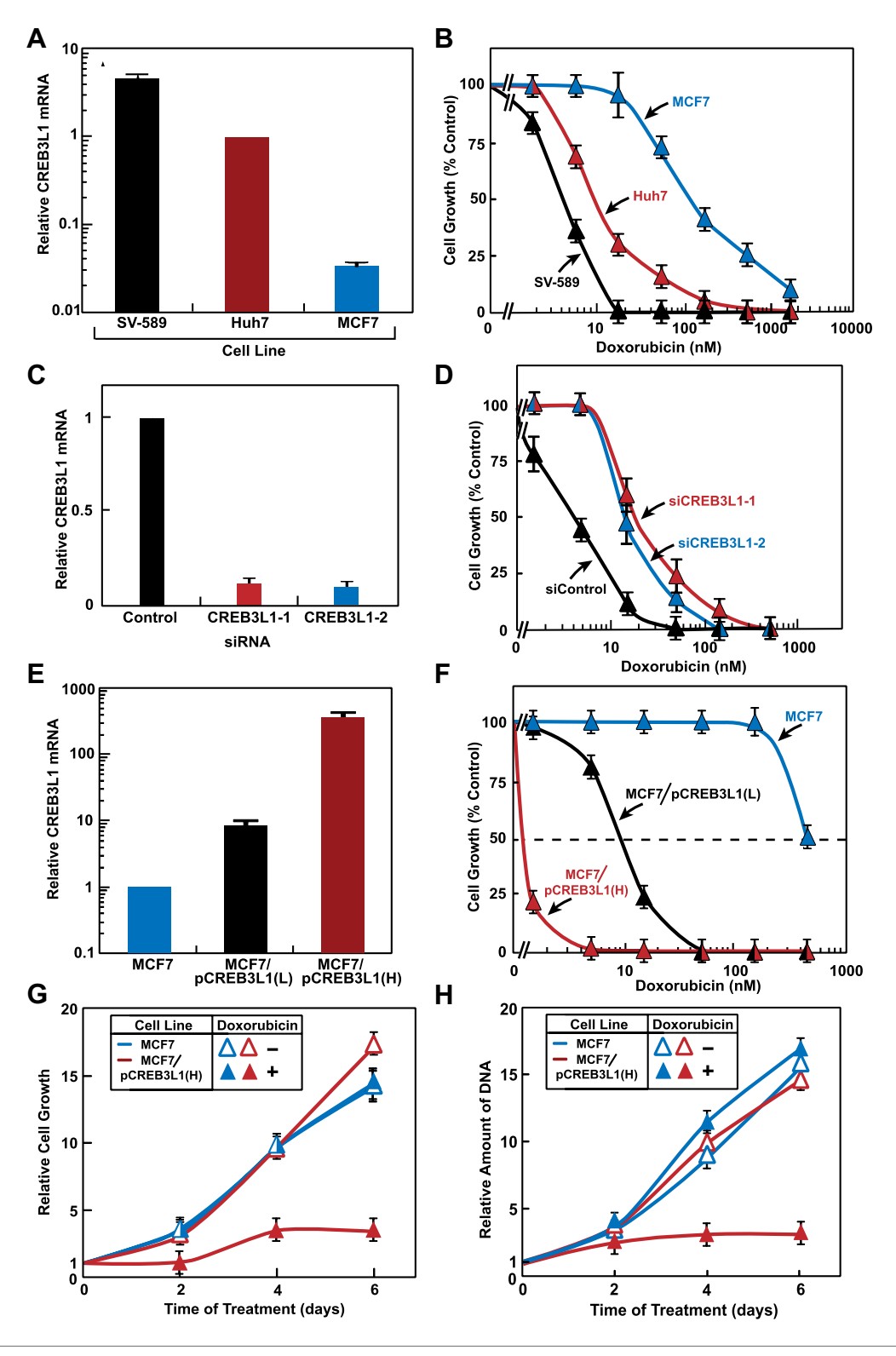

**Figure 4**. Sensitivity of cancer cells to doxorubicin is correlated to their expression of CREB3L1. (**A**),(**E**) RT-QPCR quantification of CREB3L1 mRNA in indicated cells with its value in Huh7 (**A**) or MCF-7 cells (**E**) set to 1. (**B**),(**F**) Effect of doxorubicin on proliferation of the indicated cells was determined as described in *Figure 3A*. (**C**),(**D**) SV-589 cells were treated and analyzed as described in *Figure 3D,E*. (**G**),(**H**) On day 0, indicated cells were seeded at 1.5 × 10⁵

*Figure 4. Continued on next page*

*Figure 4. Continued*

cells per 60 mm dish. On day 1, the cells were treated with or without 15 nM doxorubicin. After incubation for the indicated period of time, cell proliferation was determined by direct counting of the cells (**G**) or by measurement of the amount of cellular DNA (**H**). (**G**),(**H**) The number of cells just before doxorubicin treatment at time 0 is set to one. (**A**–**H**) Results are reported as mean ± S.E.M. of three independent experiments.

selected one clone of the cells with relatively low expression (MCF7/pCREB3L1(L); eightfold above parental cells) and another clone with high expression of CREB3L1 (MCF7/pCREB3L1(H); 300-fold above parental cells) (*Figure 4E*). The eightfold overexpression of CREB3L1 in MCF7/pCREB3L1(L) cells lowered the $IC_{50}$ for doxorubicin from 500 nM to 10 nM, and the 300-fold overexpression of CREB3L1 in MCF7/pCREB3L1(H) cells further reduced the $IC_{50}$ to ~1 nM (*Figure 4F*). In this experiment, cells were treated with doxorubicin for 2 days. To determine the effect of CREB3L1 expression on proliferation of the cells treated with doxorubicin for a longer period of time, we incubated MCF-7 and MCF7/pCREB3L1(H) cells with 15 nM doxorubicin for 6 days. This treatment did not affect proliferation of MCF-7 cells, but markedly blocked proliferation of MCF7/pCREB3L1(H) cells, as determined by direct cell counting (*Figure 4G*) and by measurement of cellular DNA content (*Figure 4H*). Thus, CREB3L1 expression level is a key determinant of cellular sensitivity to doxorubicin.

We then determined the relationship between doxorubicin-induced cleavage of CREB3L1 and DNA breaks caused by inhibition of topoisomerase. Doxorubicin induced appearance of histone γH2AX, a marker for DNA breaks (*Figure 5A*, lane 2). However, this effect was unaffected by knockdown of CREB3L1 expression (*Figure 5A*, lane 5). This result suggests that cleavage of CREB3L1 does not lead to doxorubicin-induced DNA breaks. To investigate whether DNA breaks may lead to cleavage of CREB3L1, we examined etoposide, another chemotherapeutic drug that inhibits topoisomerase (*Stähelin and von Wartburg, 1991*). Unlike doxorubicin, etoposide failed to induce cleavage of CREB3L1 (*Figure 5B*, lane 3), even though etoposide was as effective as doxorubicin in causing DNA breaks (*Figure 5A*, lanes 2 and 3). Accordingly, knockdown of CREB3L1 in Huh7 cells did not increase their resistance to etoposide (*Figure 5C*), and overexpression of CREB3L1 in MCF7 cells also did not increase their sensitivity to the compound (*Figure 5D*). These results suggest that induction of CREB3L1 cleavage by doxorubicin is not related to its inhibitory activity towards topoisomerase. Besides etoposide, CREB3L1 was also not required for bleomycin or paclitaxel to inhibit cell growth, an observation suggesting that CREB3L1 may be specifically involved in doxorubicin-induced suppression of cell proliferation (*Figure 5E–G*).

RIP of membrane-bound transcription factors is known to be a signal transduction pathway that transfers signals from the ER to nucleus (*Brown et al., 2000*). Since ER is the site where most lipids are synthesized, we wondered whether doxorubicin may alter homeostasis of certain lipids that may result in cleavage of CREB3L1. Doxorubicin and daunorubicin, a chemotherapeutic drug derived from doxorubicin, were reported to induce de novo synthesis of ceramide (*Bose et al., 1995*; *Liu et al., 2008*), a class of lipid known to inhibit cell proliferation (*Ogretmen and Hannun, 2004*). De novo synthesis of ceramide is initiated with the condensation of palmitate and serine (*Gault et al., 2010*). This rate-limiting step in de novo synthesis of ceramide is catalyzed by serine palmitoyltransferase (SPT) (*Linn et al., 2001*). Ceramide synthesis also requires a reaction catalyzed by ceramide synthase (*Mullen et al., 2012*). We confirmed that doxorubicin stimulated ceramide synthesis by showing that treatment with the compound increased the amount of [$^{14}$C]palmitate incorporated into ceramide in Huh7 cells (*Figure 6A*). Mass spectroscopy analysis revealed that doxorubicin primarily increased the amount of ceramide containing palmitate (16:0) as the amide-linked fatty acid (*Figure 6B*). In contrast to doxorubicin, etoposide failed to induce ceramide synthesis at a concentration at which cell proliferation was inhibited (*Figure 6C*).

To determine whether doxorubicin-induced ceramide synthesis is required to stimulate cleavage of CREB3L1, we treated Huh7 cells with myriocin, an inhibitor of SPT (*Miyake et al., 1995*). This treatment inhibited doxorubicin-induced cleavage of CREB3L1 (*Figure 7A*). Co-treatment with myriocin rendered the cells more resistant to doxorubicin (*Figure 7B*). Fumonisin B$_1$, an inhibitor of ceramide synthase (*Wang et al., 1991*), also blocked doxorubicin-induced cleavage of CREB3L1 (*Figure 7C*). The observations that inhibition of two different enzymes involved in ceramide synthesis were both effective in

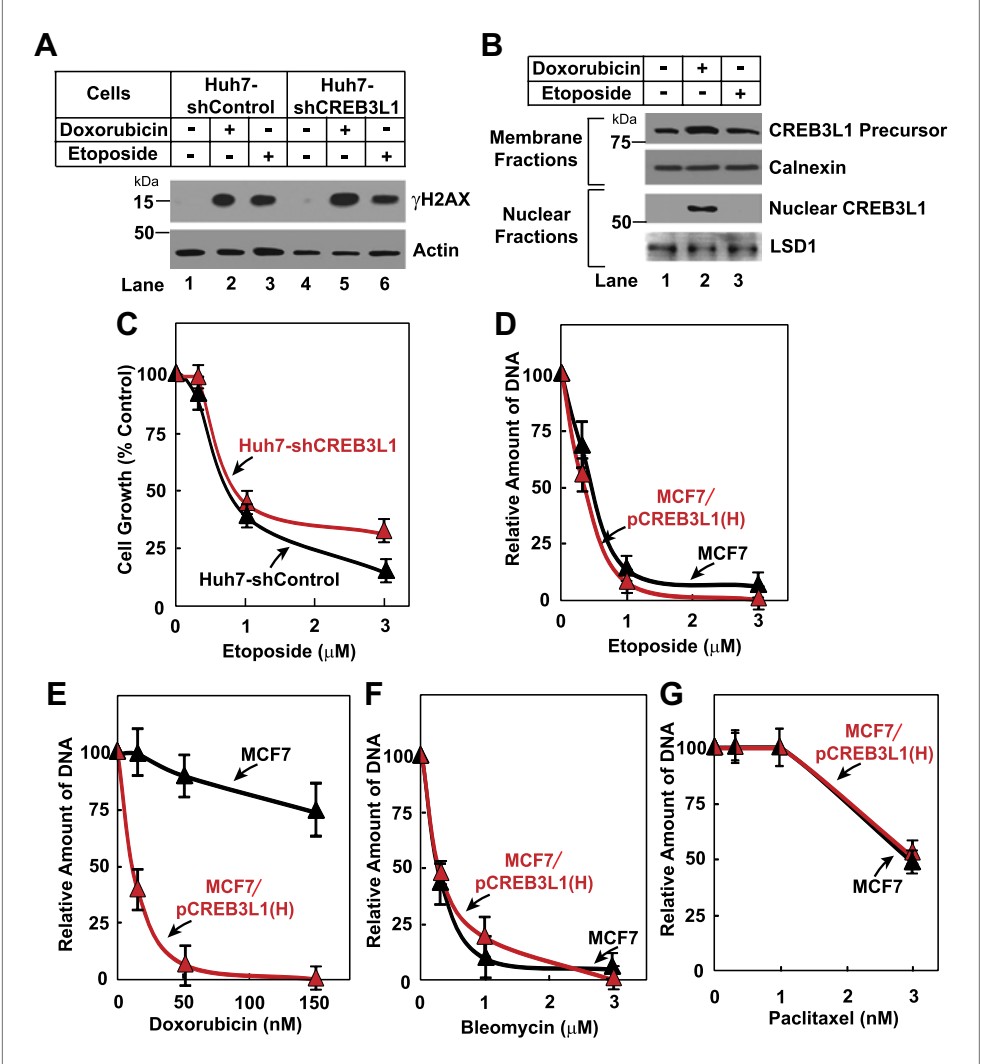

**Figure 5**. CREB3L1 activation is independent from DNA breaks. (**A**) On day 0, indicated cells were seeded at $4 \times 10^5$ cells per 60 mm dish. On day 1, cells were treated with 500 nM doxorubicin or 500 nM etoposide. On day 2, 24 hr after the treatment, the cells were harvested for immunoblot analysis with antibodies reacting against γH2AX or actin. (**B**) Huh7 cells were seeded and treated as described in (**A**). On day 2, cells were separated into nuclear and membrane fractions and analyzed by immunoblot analysis as described in *Figure 1B*. (**C**) The effect of etoposide on proliferation of the indicated cells was determined as described in *Figure 3A*. (**D**)–(**G**) On day 0, indicated cells were seeded at $1.5 \times 10^5$ cells per 60 mm dish. On day 1, cells were treated with indicated concentrations of etoposide (**D**), doxorubicin (**E**), bleomycin (**F**), or paclitaxel (**G**). On day 3, 48 hr after the treatment, proliferation of the cells was determined by measurement of cellular DNA. The amount of DNA just prior to the drug treatment and after treatment with no drug for 48 hr is set to 0% and 100%, respectively. (**C**)–(**G**) Results are reported as mean ± S.E.M. of three independent experiments.

blocking doxorubicin-induced cleavage of CREB3L1 strongly suggest that this cleavage is caused by increased synthesis of ceramide.

To more directly determine the effect of ceramide on cleavage of CREB3L1, we treated Huh7 cells with $C_6$-ceramide, a cell-permeable analogue of ceramide that contains a short acyl chain. It was reported previously that $C_6$-ceramide was converted to naturally-existing ceramide in cells through the ceramide salvage pathway (*Kitatani et al., 2008*). Indeed, our mass spectroscopy analysis confirmed that treatment with $C_6$-ceramide increased nearly all species of ceramide in Huh7 cells (*Figure 7D*). This treatment stimulated CREB3L1 cleavage even in the absence of

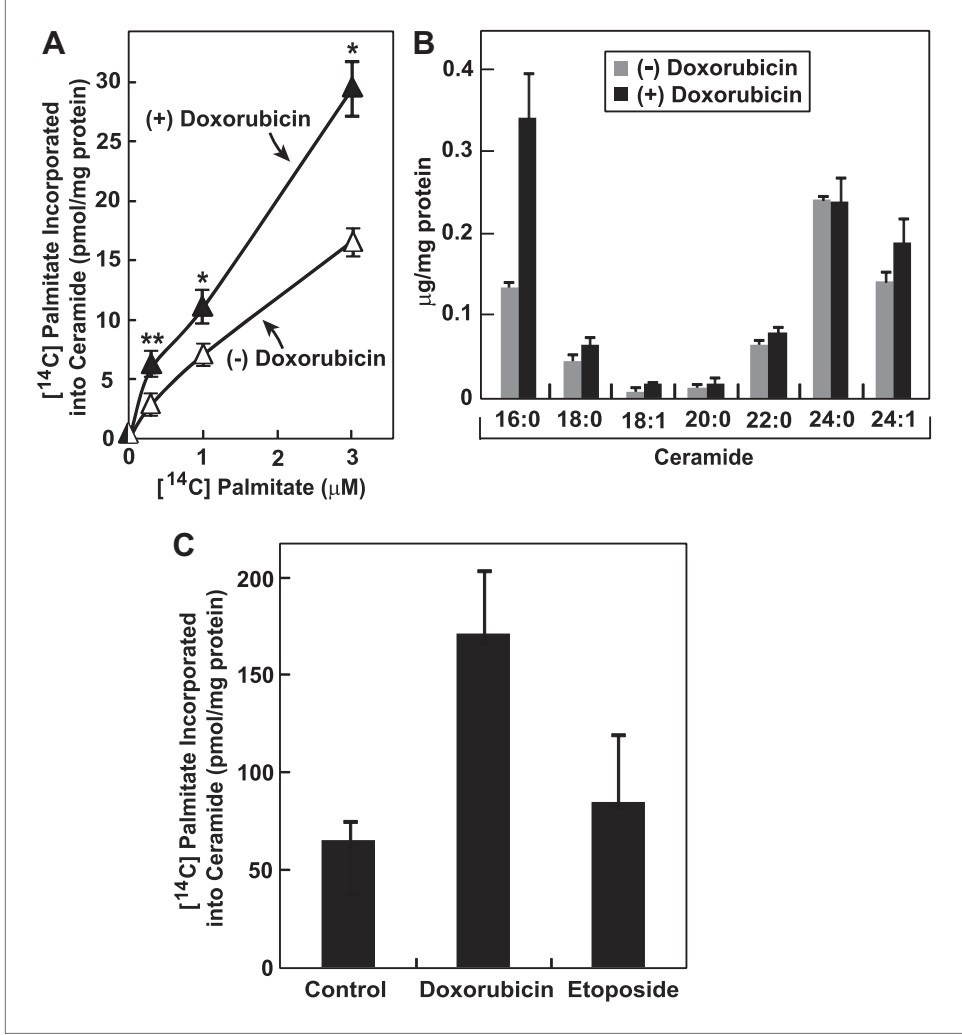

**Figure 6**. Doxorubicin stimulates synthesis of ceramide. (**A**) On day 0, Huh7 cells were seeded at $2 \times 10^5$ per 60-mm dish. On day 1, the cells were treated with or without 500 nM doxorubicin. On day 2, 20 hr after the treatment, the cells were labeled with indicated concentrations of [$^{14}$C]palmitate for additional 4 hr. Cell lipids were then extracted to determine the amount of [$^{14}$C]palmitate incorporated into ceramide. *p=0.003; **p=0.02. (**B**) On day 0, Huh7 cells were seeded at $1.5 \times 10^5$ per 60-mm dish. On day 1, the cells were treated with or without 500 nM doxorubicin. On day 2, 24 hr after the treatment, the cells were harvested for ceramide analysis via LC-MS as described in 'Materials and methods'. The amount of ceramide with indicated amide-linked fatty acids was presented. (**C**) Huh7 cells were treated with 500 nM doxorubicin or 1 μM etoposide, labeled with 3 μM [$^{14}$C]palmitate, and analyzed as described in Figure 6A. (**A**)–(**C**) Results are reported as mean ± S.E.M. of triplicate incubations from a representative experiment.

doxorubicin (*Figure 7E*). These results suggest that doxorubicin-induced synthesis of ceramide leads to cleavage of CREB3L1. Thus, CREB3L1 appears to suppress cell proliferation in response to accumulation of ceramide. This conclusion was further supported by the observation that knockdown of CREB3L1 in Huh7 cells completely abolished the ability of C$_6$-ceramide to inhibit cell proliferation (*Figure 7F*).

## Discussion

The current study establishes a crucial role for CREB3L1 in inhibiting cell proliferation in response to doxorubicin. We show that the sensitivity of cellular response to doxorubicin is positively correlated to CREB3L1 expression in cancer cells. Importantly, the concentration of doxorubicin required to

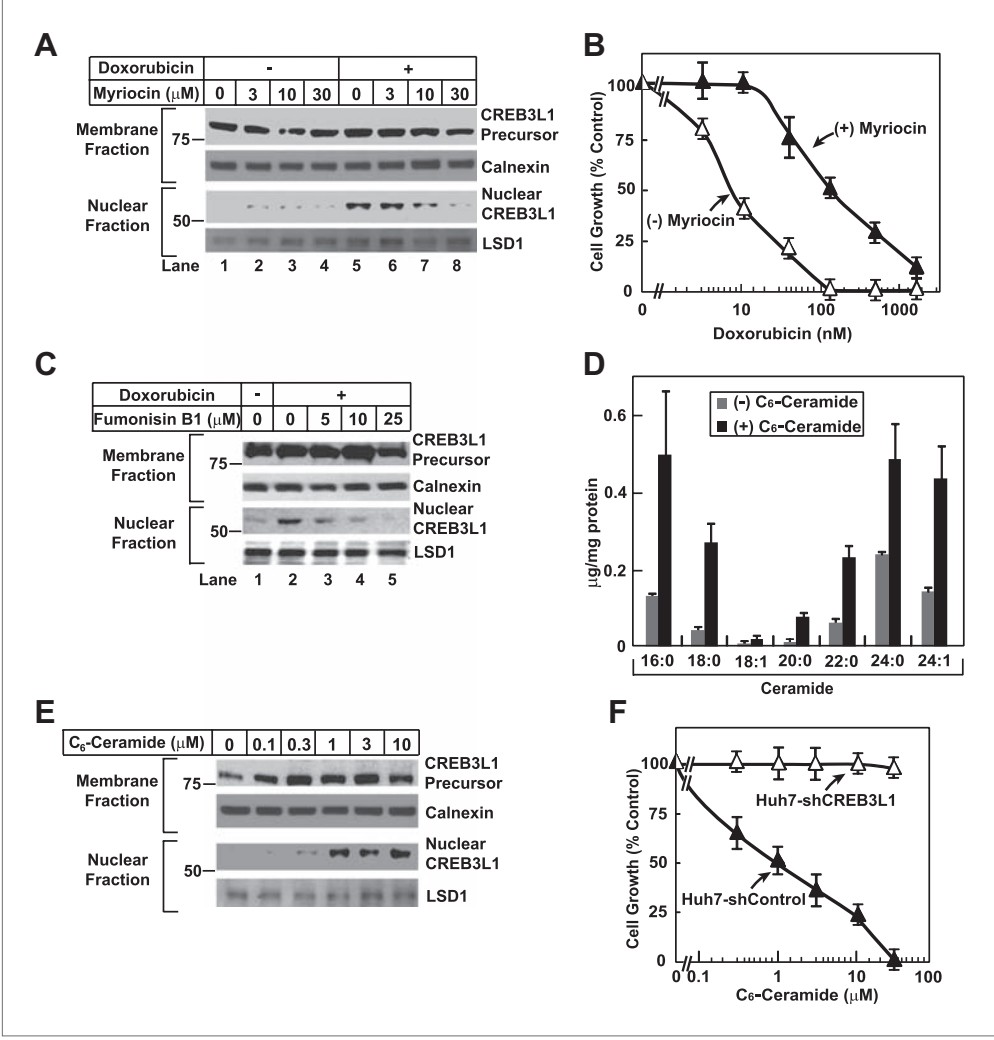

**Figure 7**. Doxorubicin-induced synthesis of ceramide stimulates cleavage of CREB3L1. (**A**),(**C**) On day 0, Huh7 cells were seeded at $4 \times 10^5$ per 60-mm dish. On day 1, the cells were treated with indicated concentrations of myriocin (**A**) or fumonisin B1 (**C**) for 2 hr, followed by co-incubation with 200 nM doxorubicin. On day 2, 24 hr after the doxorubicin treatment, cells were analyzed for cleavage of CREB3L1 by immunoblot analysis as described in *Figure 1B*. (**B**) Huh7 cells treated with or without 30 μM myriocin for 2 hr followed by co-treatment with doxorubicin were analyzed as described in *Figure 3A*. (**D**) Huh7 cells treated with 10 μM $C_6$-ceramide for 3 hr were analyzed as described in *Figure 6B*. Results are reported as mean ± S.E.M. of triplicate incubations from a representative experiment. (**E**) Huh7 cells treated with indicated concentration of $C_6$-ceramide for 24 hr were analyzed as described in *Figure 1B*. (**F**) Indicated cells treated with indicated concentration of $C_6$-ceramide for 48 hr were analyzed as described in *Figure 3A*. (**B**),(**F**) Results are reported as mean ± S.E.M. of three independent experiments.

proteolytically activate CREB3L1 is within clinically relevant concentration ranges found in the serum of patients treated with the drug (<1 μM) (*Gewirtz, 1999*). These findings raise the possibility that the clinical response to doxorubicin may be determined by the level of CREB3L1 produced in tumor cells. Thus, measuring CREB3L1 expression in tumor cells may be useful in identifying cancer patients who are most likely to benefit from doxorubicin treatment. However, this hypothesis is difficult to test with the currently available clinical data. This is because most cancer patients are treated with chemotherapy regime containing doxorubicin but not doxorubicin alone. Thus, even if the patients respond to the treatment, it is difficult to discern whether they respond to doxorubicin or other anti-cancer drugs in the regime. A clinical study using doxorubicin alone to treat tumors that express high amount of

CREB3L1 will be required to determine whether CREB3L1 expression can be used as a biomarker to predict treatment outcome of doxorubicin.

An important finding in the current study is that doxorubicin-induced accumulation of ceramide is required for cleavage of CREB3L1. This is the second example of a transcription factor whose proteolytic activation is regulated by a lipid synthesized in the ER. The first such example is SREBP-2, a transcription factor that regulates cholesterol metabolism (*Brown and Goldstein, 2009*). When ER cholesterol content is less than 4% of total lipid, SREBP-2 are transported from the ER to Golgi complex where it is cleaved by S1P and S2P (*DeBose-Boyd et al., 1999*; *Radhakrishnan et al., 2008*). These cleavages liberate the $NH_2$-terminal domain of SREBP-2 from membranes, allowing it to enter the nucleus where it activates all genes required for cholesterol synthesis and uptake (*Horton et al., 2003*). When ER cholesterol content exceeds 8% of total lipid, SREBP-2 is retained in the ER so that it is separated from S1P and S2P that are localized in the Golgi complex. Consequently, cleavage of SREBP-2 is inhibited (*Nohturfft et al., 2000*; *Radhakrishnan et al., 2008*). If the mechanism through which ceramide regulates cleavage of CREB3L1 is similar to that employed by cholesterol to regulate cleavage of SREBP-2, then excessive ceramide is predicted to trigger the transportation of CREB3L1 from the ER to Golgi complex. Such similarity might also explain why a twofold increase in ceramide is sufficient to induce cleavage of CREB3L1, as ceramide may also function through the same switch-like mechanism used by cholesterol to regulate SREBP-2 cleavage.

Our current study demonstrates that proteolytic activation of CREB3L1 is required for doxorubicin to induce expression of *p21*. However, expression of *p21* alone may not be sufficient to suppress cell proliferation. We have shown previously that CREB3L1 induces transcription of multiple genes that suppress cell proliferation (*Denard et al., 2011*). Thus, CREB3L1 may function similar to p53 as a master regulator of cell proliferation. It was reported previously that doxorubicin inhibited cell proliferation through both p53 dependent and independent pathways (*Lupi et al., 2007*). Since CREB3L1 is able to inhibit proliferation of doxorubicin-treated Huh7 cells in which p53 is inactivated by mutations (*Hsu et al., 1993*), CREB3L1-mediated pathway is likely to be p53-independent. Most cancer cells are thought to originate from genome damage. Since p53 is activated in response to genome damage to inhibit cell proliferation, the protein is frequently inactivated by mutations in human cancer cells (*Levine et al., 1991*). Unlike p53, CREB3L1 is activated by ceramide or ER stress (*Murakami et al., 2006*, *2009*) but not genome damage. Owing to the lack of selection pressure against expression of CREB3L1, most cancer cells may still express functional CREB3L1. This may be the reason why doxorubicin is effective against many varieties of cancers. Thus, more effective chemotherapeutic reagents against cancers may be generated by development of compounds that specifically activate CREB3L1.

## Materials and methods

### Materials

We obtained rabbit anti-LSD1 from Cell Signaling (Boston, MA); mouse anti-calnexin from Enzo Life Sciences (Farmingdale, NY); mouse anti-γH2AX from Millipore (Billerica, MA); rabbit anti-Actin and anti-p21 from Abcam (Cambridge, MA); peroxidase-conjugated secondary antibodies from Jackson ImmunoResearch (West Grove, PA); Doxorubicin (Cat# D1515-10MG), bleomycin, etoposide, paclitaxel and N-Hexanoyl-D-sphingosine ($C_6$-Ceramide) from Sigma-Aldrich (St. Louis, MO); Myriocin and fumonisin $B_1$ from EMD Biosciences (Darmstadt, Germany); and [$^{14}$C]palmitate (55 mCi/mmol) from ARC (St. Louis, MO). A rabbit polyclonal antibody against human CREB3L1 was generated as previously described (*Denard et al., 2011*). Doxorubicin stock solution (2.5 mg/ml) was made by adding nuclease-free water (Ambion, Carlsbad, CA) directly to the vial, and was stored at 4°C for no more than 2 weeks.

### Cell culture

SRD-12B and M19 cells are mutant CHO cells deficient in S1P and S2P, respectively (*Rawson et al., 1997*, *1998*). These cells were maintained in medium A (1:1 mixture of Ham's F12 medium and Dulbecco's modified Eagle's medium containing 100 U/ml penicillin and 100 μg/ml streptomycin sulfate) supplemented with 5% (vol/vol) fetal calf serum (FCS), 5 μg/ml cholesterol, 1 mM sodium mevalonate, and 20 μM sodium oleate. Their parental CHO-7 cells are a clone of CHO-K1 cells selected for growth in lipoprotein-deficient serum (*Metherall et al., 1989*) and were maintained in medium A supplemented with 5% (vol/vol) newborn calf lipoprotein-deficient serum. Huh7 and SV589 cells were maintained in medium B (Dulbecco's modified Eagle's medium with 4.5 g/l glucose, 100 U/ml

penicillin, 100 mg/ml streptomycin sulfate, and 10% [vol/vol] FCS). Single cell clones of Huh7-shControl and Huh7-shCREB3L1 cells were generated by stably transfecting Huh7 cells with a control shRNA or shRNA targeting CREB3L1, respectively, as previously described (*Denard et al., 2011*). These cells were maintained in medium B supplemented with 10 µg/ml puromycin. MCF-7 cells were maintained in medium C (RPMI-40 media with 100 U/ml penicillin, 100 mg/ml streptomycin sulfate, and 10% [vol/vol] FCS). MCF7/pCREB3L1(L) and MCF7/pCREB3L1(H) were generated by stably transfecting MCF-7 cells with pTK-CREB3L1 encoding human CREB3L1 driven by the thymidine kinase promoter. These cells were maintained in medium C supplemented with 700 µg/ml G418. All cells were incubated in monolayers at 37°C in 5% $CO_2$ except for CHO and MCF-7-derived cells that were cultured at 37°C in 8% $CO_2$. None of the cells were allowed to reach more than 80% confluence during maintenance.

## Transfection

Cells were transfected with indicated plasmids using Fugene 6 reagent (Promega) as described by the manufacturer, after which the cells were used for experiments as described in the 'Figure legends'.

## Immunoblot analyses

Cell homogenates were separated into nuclear and membrane fractions (*Sakai et al., 1996*), and analyzed by SDS-PAGE (15% for γH2AX, and 10% for the rest of the proteins) followed by immunoblot analysis with the indicated antibodies (1:1000 dilution for anti-CREB3L1 and anti-LSD1, 1:3000 dilution for anti-calnexin, 1:2000 dilution for anti-γH2AX and 1:10,000 dilution for anti-Actin). Bound antibodies were visualized with a peroxidase-conjugated secondary antibody using the SuperSignal ECL-HRP substrate system (Pierce).

## RT-QPCR

RT-QPCR was performed as previously described (*Liang et al., 2002*). Each measurement was made in triplicate from cell extracts pooled from duplicate dishes. The relative amounts of RNAs were calculated through the comparative cycle threshold method by using human 36B4 mRNA as the invariant control.

## Cell quantification

The number of cells was determined by direct counting or measurement of cellular DNA content with Quant-iT dsDNA Assay Kit (Life Technologies). Results from each experiment were reported as the mean value from triplicate incubations.

## RNA interference

Duplexes of siRNA were synthesized by Dharmacon Research. The siRNA sequences targeting human CREB3L1 and the control siRNA targeting GFP was reported previously (*Adams et al., 2004*; *Denard et al., 2011*). Cells were transfected with siRNA using Lipofectamine RNAiMAX reagent (Invitrogen) as described by the manufacturer, after which the cells were used for experiments as described in the figure legends.

## TUNEL assay

TUNEL assay was performed with the APO-BrdU TUNEL Assay kit (Invitrogen) as described in the manufacturer's directions. Cells were subjected to flow cytometry on a FACSCaliber Flow Cytometer (Becton Dickinson) to determine percent of apoptotic cells. At least 5000 cells were collected for each measurement. Results from each experiment were reported as mean of triplicate measurements.

## Measurement of ceramide synthesis

Ceramide synthesis measured by radiolabeled analysis was performed by incubating cells with [$^{14}$C] palmitate followed by homogenizing the cells in buffer A (10 mM HEPES pH 7.6, 1.5 mM $MgCl_2$, and 10 mM KCl). Lipids in the homogenate were extracted by 0.5 ml of chloroform/methanol (2:1; vol/vol), dried, and dissolved in 70 µl of chloroform/methanol (1:1; vol/vol). Lipid extracts were mixed with 50 µg of non-radioactive ceramide standard (Avanti Polar Lipids) and analyzed by Thin Layer Chromatography (TLC) on POLYGRAM SIL G plates in a solvent system of chloroform/acetate (90:10; vol/vol) for ceramide separation. Following visualization by exposing the TLC plates to $I_2$ vapor, bands containing ceramide were excised, and the amount of radioactivity in it was determined by scintillation counting. The activity of ceramide synthesis was determined by radioactivity found in the ceramide band normalized by the amount of cellular protein. The statistical analysis was performed with one tailed paired t-test.

LC-MS analyses of ceramide were performed by UPLC-MS/MS at UT Southwestern Medical Center Mouse Metabolic Phenotyping Core. The equipment consisted of a Shimadzu Prominence UPLC system equipped with a CBM-20A controller, a DGU-A3 degasser, three UPLC solvent delivery modules LC-ADXR, a CTO-20AC column oven/chiller maintained at 30°C, a SIL-20ACTHT autosampler. The UPLC system is attached to an API 5000 LC-MS/MS system (Applied Biosystems/MDS SCIEX, Concord ON, Canada). The mass spectrometer is equipped with a Turbo V ion source operating the TurboIonSpray probe in positive mode. Quantitative analysis of sphingolipids was achieved using selective reaction monitoring scan mode. Chromatographic separations were obtained by reverse phase LC on a 2.1 (i.d.) × 150 mm Kinetex C8 (Phenomenex, Torrance, CA) column under a complex gradient elution, using three different mobile phases: eluent A consisting of $CH_3OH/H_2O/HCOOH$, 58/41/1, vol/vol/vol with 5 mM ammonium formate, eluent B consisting of $CH_3OH/HCOOH$, 99/1, vol/vol with 5 mM ammonium formate, and eluent C consisting of $CH_3OH/CH_2Cl_2$ 35/65 with 5 mM ammonium formate. The amount of ceramide measured was normalized against the amount of cellular protein.

## Acknowledgements

We thank Drs Michael S Brown and Joseph L Goldstein for their support and advice; Lisa Beatty, Shomanike Head and Ijeoma Onwuneme for help with tissue culture; Jeff Cormier for help with RT-QPCR; and Saada Abdalla and Kristina Garner for technical assistance.

## Additional information

### Funding

| Funder | Grant reference number | Author |
| --- | --- | --- |
| National Institutes of Health | AI 090119 | Jin Ye |
| National Institutes of Health | HL 20948 | Jin Ye |

The funder had no role in study design, data collection and interpretation, or the decision to submit the work for publication.

### Author contributions

BD, Conception and design, Acquisition of data, Analysis and interpretation of data, Drafting or revising the article; CL, Acquisition of data, Analysis and interpretation of data; JY, Conception and design, Analysis and interpretation of data, Drafting or revising the article.

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
