## [Author Response]

Thank you very much for your evaluation of our manuscript. The constructive criticism is greatly appreciated. We revised our manuscript according to your suggestions. We hope that the revised manuscript is now suitable for publication. The detailed revision is listed as follows.

*1. The authors need additional evidence that the effects of doxorubicin are mediated through an increase in ceramide levels before this can be published*.

We now include three more experiments to show that doxorubicin stimulates CREB3L1 cleavage through increased synthesis of ceramide. In addition to the radiolabeling experiment presented in the original manuscript, we used mass spectroscopic analysis to demonstrate that doxorubicin increased intracellular amounts of ceramide. This new result is presented in Figure 6B. We also show that treatment with fumonisin B1, an inhibitor of ceramide synthase, also blocked doxorubicin-induced cleavage of CREB3L1. This result is presented in Figure 7C. The observations that two inhibitors (myriosin and fumonisin B1) targeting two different enzymes involved in ceramide synthesis were both effective in inhibiting doxorubicin-induced cleavage of CREB3L1 strongly suggest that increased ceramide synthesis led to this cleavage.

Finally, through mass spectroscopic analysis, we show that treatment with C6-ceramide increased nearly all species of naturally existing ceramide in cells. This result is presented in Figure 7D. This observation confirms earlier reports that C6-ceramide functions through increasing intracellular amount of naturally existing ceramide, as opposed to acting as a mimetic for natural long chain ceramide.

[Comment 2 no longer applies.]

*3. The authors do not examine the effects of etoposide on ceramide*.

We now show that etoposide failed to induce ceramide synthesis at a concentration at which cell proliferation was inhibited. This result is presented in Figure 6C.

*4. The shRNA studies are based on the use of a single shRNA*.

For reasons that we do not understand, after screening for nearly 100 clones of Huh7 cells transfected with four different shRNA targeting CREB3L1, we only obtain a single clone of the cells in which CREB3L1 is consistently knocked down by more than 90%. We were unable to perform the rescue study because overexpression of CREB3L1 led to constitutive cleavage of the protein that blocked proliferation of Huh7 cells even in the absence of doxorubicin. We thus knocked down CREB3L1 by transiently transfecting cells with two distinct siRNA targeting CREB3L1 at regions different from that targeted by the shRNA. Transfection of these siRNA also made Huh7 cells more resistant to doxorubicin. These new results, which are presented in Figures 3D and 3E should rule out the off target effect of the shRNA.

*5. The relevance of p53 status to the findings reported is unclear*.

Our finding that CREB3L1 is able to inhibit proliferation of Huh7 cells in which p53 is inactivated by mutations indicates that CREB3L1-mediated pathway is p53 independent.

This point is discussed in more detail in the revised manuscript. Unfortunately, we were unable to perform the experiment suggested by the Senior Editor because doxorubicin-triggered CREB3L1-dependent responses only occur at low concentrations of doxorubicin that do not result in cell apoptosis.

*6. Inclusion of Oncomine data*.

We already discussed the reason why we were unable to obtain patient-related data from the database in our original manuscript. The database does allow comparison in gene expression profiles between cultured cancer cells that are doxorubicin resistant and those that are doxorubicin sensitive. However, the concentrations of doxorubicin used in the studies were not described. Traditionally, these analyses were performed with doxorubicin at μM ranges to study the function of multidrug resistance gene. At these high concentrations, doxorubicin induces cellular toxicity independent of CREB3L1.

Thus, such comparison does not generate useful data for us to determine the role of CREB3L1 in doxorubicin-mediated suppression of cell proliferation.